# PRAME Promotes Cervical Cancer Proliferation and Migration via Wnt/β-Catenin Pathway Regulation

**DOI:** 10.3390/cancers15061801

**Published:** 2023-03-16

**Authors:** Xin Chen, Mengying Jiang, Shengjie Zhou, Hong Chen, Gendi Song, Yichen Wu, Xueqiong Zhu

**Affiliations:** 1Department of Obstetrics and Gynecology, The Second Affiliated Hospital of Wenzhou Medical University, Wenzhou 325027, China; 2Department of Obstetrics and Gynecology, Taizhou Women and Children’s Hospital of Wenzhou Medical University, Taizhou 318000, China

**Keywords:** PRAME, cervical cancer, Wnt/β-catenin signaling, tumorigenesis, proliferation

## Abstract

**Simple Summary:**

Preferentially expressed antigen in melanoma (PRAME), a member of the CTA gene family, was first reported as a cancer/testis antigen. In this research, PRAME was found to be highly expressed in cervical cancer tissues and cells compared with control groups. PRAME-knockdown and -overexpression cell models were constructed. Our results demonstrated that PRAME promoted cell proliferation, migration, and invasion and reduced cell apoptosis and G0/G1 arrest by activating the Wnt/β-catenin pathway. There may be implications for future cervical cancer treatments based on these findings.

**Abstract:**

A significant burden is placed on the lives of females due to cervical cancer, which is currently the leading cause of cancer death among women. Preferentially expressed antigen in melanoma (PRAME) belongs to the CTA gene family and was found to be abnormally expressed among different types of cancers. Our previous research also indicated that PRAME was highly expressed in cervical cancer compared with normal tissues. However, the roles and detailed mechanisms of PRAME have not been explored in cervical cancer. In the present study, the expression of PRAME in cervical tissues and cells was detected by immunohistochemistry (IHC), qRT-PCR, and Western blotting. Additionally, CCK-8, BrdU, scratch, transwell, and flow cytometry assays were conducted to explore the function of PRAME in regulating the malignant biological behaviors of cervical cancer cells. Nude mice were used to confirm the role of PRAME in tumor growth in vivo. Furthermore, the Wnt inhibitor MSAB was used to verify the role of PRAME in regulating the Wnt/β-catenin pathway both in vitro and in vivo. The results of IHC, qRT-PCR, and Western blotting showed that PRAME was highly expressed in cervical cancer tissues and cells. PRAME knockdown attenuated cell growth, migration, and invasion; induced G0/G1 arrest; and increased cell apoptosis in C33A and SiHa cells through Wnt/β-catenin signaling regulation. However, the upregulation of PRAME exhibited the opposite effects accordingly, which could be partly reversed via MSAB treatment. The growth rate of xenograft tumors was enhanced when PRAME was overexpressed via Wnt/β-catenin signaling activation. Taken together, PRAME is associated with cervical cancer occurrence and progression mediated by Wnt/β-catenin signaling, suggesting that PRAME might be a factor in manipulating cervical carcinogenesis and a potential therapeutic target.

## 1. Introduction

As the fourth most common cancer worldwide, cervical cancer poses a significant burden on the lives of women, despite advances in cervical cancer vaccines and other screening and diagnostic technologies [1]. The latest survey estimated that there were more than 410,000 new cases and 97,000 deaths worldwide annually [1]. Cervical squamous cancer has been associated with the persistent infection of high-risk HPV [2]. Since the implementation of vaccination is uncoordinated between countries with different economic incomes, cervical cancer will remain a significant global health burden for some time to come [1]. Resistance to chemotherapy and radiotherapy often leads to unsatisfactory treatment outcomes for advanced and recurrent cervical cancer. Additionally, tumor metastasis also leads to a decline in the survival rate of cervical cancer patients. Therefore, it is urgent for us to identify key targets that can predict the occurrence and metastasis of cervical cancer early, providing novel strategies for cervical cancer therapy in the future.

Preferentially expressed antigen in melanoma (PRAME), a coding human leucocyte antigen (HLA)-A24, was first reported as a cancer/testis antigen in 1997 [3]. PRAME has been considered to be a significant target in immunotherapy due to its restricted re-expression in cancer [4]. Cancer vaccines and adoptive T cell therapies have been developed based on the strong specific immune responses against tumor cells elicited by PRAME [5,6].

Multiple studies have shown that the expression of PRAME is abnormal in different tumors, including melanoma [7], hematological malignancies [8], breast cancer [9], lung cancer [10], and sarcoma [11]. However, it is almost undetectable in normal human tissues [12]. One study found that high expression of PRAME inhibited tumor metastasis in non-small cell lung cancer [13], and some bioinformatics analyses suggested that PRAME overexpression was associated with a better prognosis [14]. However, AlKhadairi et al. [9] found that PRAME expression promoted Epithelial–Mesenchymal Transition (EMT) in triple-negative breast cancer. PRAME was also found to be involved in the progression of liver cancer and lung cancer through cell cycle regulation via the degradation of p14/ARF [15]. The effect of PRAME on tumor promotion or inhibition depends on tumor types. In our previous study, PRAME was differentially expressed in cervical cancer and normal tissues [16]. Nonetheless, the role of PRAME in cervical cancer occurrence and progression, as well as the underlying mechanism, is obscure. This study was designed to provide more evidence on the function of PRAME in the tumorigenesis of cervical cancer.

## 2. Materials and Methods

### 2.1. Cell Lines and Reagents

All cell lines of cervical cancer (SiHa, HeLa, CaSki, and C33A) and normal cervical squamous epithelium (Ect1/E6E7) were cultured with basal medium containing 10% fetal bovine serum (PAN-Biotech, Aidenbach, Germany) and 1% penicillin–streptomycin solution (Meiluncell, Dalian, China) in a 37 °C humidified incubator under 5% CO_2_. Dulbecco’s Modified Eagle Medium (Meilunbio, Dalian, China) was used to culture SiHa, HeLa, and C33A cells. Roswell Park Memorial Institute 1640 medium (Gibco, Waltham, MA, USA) was used to culture CaSki cells. Eagle’s Minimum Essential Medium (Hyclone, Logan, UT, USA) was used to culture Ect1/E6E7 cells. MSAB (MedChemExpress, Monmouth Junction, NJ, USA) is a potent inhibitor of Wnt/β-catenin signaling which can downregulate target genes of the Wnt/β-catenin pathway. MSAB was dissolved in DMSO at a concentration of 50 mM. A final concentration of 5 μM of MSAB was used in vitro, and a 15 mg/kg concentration was used in vivo.

### 2.2. Immunohistochemistry (IHC)

Ethics approval was obtained from the Second Affiliated Hospital of Wenzhou Medical University regarding the procedure related to human subjects. Twenty-two cases of cervical cancer tissues and adjacent normal cervical tissues embedded in paraffin were obtained from patients diagnosed and pathologically confirmed in the Second Affiliated Hospital of Wenzhou Medical University from 2019 to 2022. The tissues were cut into slices with a 4 μm thickness. The paraffin slices of these tissues were placed in a drying oven at 60 °C for 2 h, then dewaxed with xylene and gradient ethanol. Antigen repair was performed by immersing the sections in sodium citrate buffer and microwaving them. The tissue slices were treated with rabbit anti-human PRAME antibody (1:200, orb373396, Biorbyt) at 4 °C overnight. The slices were then incubated with the corresponding anti-rabbit IgG (MXB UltraSensitiveTM SP IHC Kit, Fuzhou, China) after the slices had been washed with PBST three times. Finally, 3,3′-diaminobenzidine was used to stain the target protein in the cytoplasm, and a hematoxylin dye solution was used to visualize the nuclei. The sample scores were judged according to the following criteria. Semi-quantitative analysis of staining results: the staining intensity score multiplied by the score of the percentage of positive cells [17]. The staining intensity score was classified into 0 (no staining), 1 (weak staining), 2 (moderate staining), and 3 (strong staining). Staining area score: 1 (<25%), 2 (26–50%), 3 (51–75%), and 4 (>75%).

### 2.3. Quantitative Real-Time PCR (qRT-PCR)

Total RNA was extracted from the cervical cancer cells and normal cervical cells with RNA-easy^TM^ Isolation Reagent (Vazyme, #R701, Nanjing, China), and the quantitative cDNA was synthesized via reverse transcription and used for qRT-PCR. The human PRAME primer was listed as follows: forward, 5′-CGTGCCTGAGCAACTGAT-3′; reverse, 5′-TACCCACCTTGGCGAAAT-3′. The expression of PRAME mRNA was normalized to β-actin expression according to the 2^−ΔΔCt^ formula. The PRAME mRNA expression in Ect1/E6E7 cells was defined as 1 for comparison.

### 2.4. Western Blotting Analysis

The collected cell samples were lysed by using RIPA lysis buffer containing PMSF (Beyotime, Shanghai, China) on ice. The precipitates were removed after centrifugation to obtain total cell protein.

The protein concentration of samples was measured via a BCA protein Assay kit (Beyotime, China), and the same protein systems (40 μg/20 μL) were prepared in a certain proportion with the loading buffer (Biosharp, Hefei, China). Target proteins were separated by 7.5%, 10%, or 12.5% SDS-PAGE gel electrophoresis at 70 V for 40 min and at 130 V for 60 min. Protein was then transferred to the PVDF membrane via a transfer tank with 300 mA for 90 min. The membranes were treated with the rabbit anti-PRAME antibody (1:1000, A14507, ABclonal, Woburn, MA, USA), rabbit anti-E-cadherin antibody (1:1000, 20874-1-AP, Proteintech, Rosemont, IL, USA), mouse anti-N-cadherin antibody (1:1000, 22018-1-AP, Proteintech), rabbit anti-Wnt3a antibody (1:1000, A0642, ABclonal), rabbit anti-Wnt5a/b antibody (1:1000, #2721, CST, Danvers, MA, USA), rabbit anti-p-LRP6 antibody (1:1000, #2568, CST), rabbit anti-β-Catenin antibody (1:1000, #8480, CST), rabbit anti-LEF1 antibody (1:1000, #2230, CST), mouse anti-CD44 antibody (1:1000, #3570, CST), rabbit anti-Cyclin D1 antibody (1:1000, #2978, CST), mouse anti-GAPDH antibody (1:5000, 60004-1-Ig, Proteintech), and rabbit anti-β-actin antibody (1:5000, 81115-1-RR, Proteintech) overnight at 4 °C. PVDF membranes were washed with TBST 5 times for 3 min, followed by 1.5 h of incubation with anti-rabbit (1:5000, BL052A, Biosharp) or anti-mouse (1:5000, BL051A, Biosharp) secondary antibodies. Residual secondary antibodies were then washed with TBST. Lastly, target protein bands were visualized by using an enhanced chemiluminescence kit (Meilunbio, China) and normalized to internal standards.

### 2.5. Plasmid and Cell Transfection

The PRAME-pCDH-GFP+Puro was ordered from Youbio Biosciences Inc. (Changsha, China), and the shRNA sequences were designed by Sigma-Aldrich (Shanghai, China). 293T cells were transfected with the target plasmids, lentivirus packaging helper plasmids, and transfection reagent Lipo2000 (Invitrogen, Waltham, MA, USA). After 2 d and 3 d, the lentiviral particles were collected from the medium of 293T cells to infect C33A and SiHa cells. The stable constructed cells were selected through puromycin (2 μg/Ml). Generally, for gene knockdown, two to four shRNA sequences were designed for each gene depending on the target site to prevent off-target effects. Two shRNA sequences were used for PRAME knockdown and are listed as follows: sh-PRAME-1, 5′- GGAAGGTGCCTGTGATGAATT -3′; sh-PRAME-2, 5′- GCTCCCAGCTTACGACCTTAA -3′.

### 2.6. CCK-8 Assay

C33A cells at a concentration of 5 × 10^3^ cells/well and SiHa cells at a concentration of 2 × 10^3^ cells/well were seeded in 96-well plates with complete medium. On day 0, 1, 2, 3, and 4, 10 Μl CCK-8 solution was added to every well. The cell viability was assessed through absorbance of 450 nm wavelength via Cytation (BioTek, Winooski, VT, USA) after incubation for 3 h in a 37 °C incubator.

### 2.7. Cell Proliferation Assay

The BrdU and colony formation assays were used to evaluate the proliferative ability of cervical cancer cells. For the BrdU assay, 7 × 10^5^/well C33A cells and 5 × 10^5^/well SiHa cells were seeded on cell-crawling slices in 6-well plates. The cell medium was added with the BrdU reagent. Then, the cells were treated with 4% tissue fixing fluid for 30 min and 0.2% Triton X-100 for 30 min after culturing for 8 h. The cell climbing slides were incubated with the mouse anti-BrdU antibody (1:200, 5292S, CST) at 4 °C overnight, and then, anti-mouse IgG/HRP staining reagent (ZSGBbio, DS-0003, Beijing, China) was used to visualize the brown staining cells under the microscope (Leica, Wetzlar, Germany). Each group was captured in 5 visual fields, and the percentage of positive cells was calculated. The experiment was repeated three times.

### 2.8. Cell Apoptosis and Cell Cycle Assay

For the cell apoptosis assay, cell samples were trypsinized and resuspended in binding buffer. The cell samples were stained with PE and 7-AAD (BD Biosciences, Franklin Lakes, NJ, USA) for cell apoptosis detection.

For the cell cycle assay, the trypsinized cells were fixed with 70% ethanol at −20 °C overnight and then treated with Propidium Iodide (PI) at 37 °C for 30 min to detect cell cycle distribution. PI was a fluorescent nucleic acid dye which could be selectively embedded between the bases of the DNA double-stranded helix. The amount of PI binding was directly proportional to the content of DNA. The DNA distribution of each stage of the cell cycle was analyzed via flow cytometry and Flowjo 10.0, and the percentage of each stage of the cell cycle was calculated.

### 2.9. Cell Migration and Invasion Assay

In order to evaluate the migratory ability, a wound-healing assay was performed in PRAME-knockdown and -overexpressed cells. A wound was scratched on the single-cell layer upon the cell reaching 90% confluence. The cells were treated with mitomycin (1 μg/mL) for 4 h and then cultured in new serum-free medium for inhibiting the effect of cell division and proliferation. At 0 h and 24 h, the wound pictures were captured via a microscope (Leica, Germany), and the area was calculated according to the method described previously [16].

Furthermore, a transwell assay was also used to evaluate the migratory and invasive abilities of cells. An amount of 5 × 10^4^ SiHa cells was seeded in a transwell chamber (Constar, Corning, NY, USA) with serum-free medium, and 8 × 10^4^ cells were seeded in a chamber with Matrigel (BD, USA) that was added in advance. After 24 h, the cells that crossed the polycarbonate membranes were stained with crystal violet, photographed, and counted in 5 visual fields by using a microscope (Leica, Germany). The cells had been treated with mitomycin (1 μg/mL) before these experiments for inhibition of cell division and proliferation, and the transwell assay was performed only on SiHa cells due to the poor migratory and invasive abilities of C33A.

### 2.10. Animal Study

BALB/c nude mice aged 4–6 weeks (Beijing Weitonglihua Sciences Co., Inc., Beijing, China) were used in this research, and the experiment was approved by the Institutional Animal Care and Use Committee of Wenzhou Medical University. PRAME-knockdown or control tumor models were established by subcutaneously injecting each nude mouse with 100 μL PBS containing 3 × 10^6^ PRAME-silenced SiHa cells or control SiHa cells, respectively. PRAME-overexpressed or control tumor models were established by injecting 3 × 10^6^ PRAME-overexpressed C33A cells or control C33A cells in the hind limb of each mouse. The tumor volume was calculated according to the formula length × width^2^/2.

In order to verify the role of the Wnt/β-catenin pathway in the PRAME-overexpressed tumor-promoting effect, the Wnt inhibitor MSAB was used in vivo. After tumor formation, the PRAME-overexpressed or control nude mice were further divided into four groups (G1: Control cDNA+DMSO, G2: PRAME cDNA+DMSO, G3: Control cDNA+MSAB, and G4: PRAME cDNA+MSAB). The mice were injected intraperitoneally with MSAB (15 mg/kg) or vehicle solution (DMSO) once a week when the tumor volume reached approximately 50 mm^3^. Tumor volume was measured and calculated regularly following MSAB therapy.

Then, paraffin sections were prepared by using xenograft tissues from sacrificed mice. The apoptosis level of tumor cells was detected by using the in situ cell death detection kit, POD (Roche, Basel, Switzerland). Immunofluorescence staining and IHC assay were used to evaluate the migratory and proliferative abilities of tumor cells. Immunofluorescence was performed as described previously [18]. Briefly, the slides were incubated with mouse anti-Ki67 antibody (1:200, 9449S, CST) at 4 °C overnight and then treated with fluorescent secondary antibodies. Cell nuclei were displayed as blue fluorescence by Hoechst 33342 staining (C1022, Beyotime). Mean fluorescence intensity analysis was used to present the level of protein expression. Moreover, the IHC staining in the slides of G1, G2, G3, and G4 mice tissues was performed to incubate with mouse anti-Ki67 antibody, rabbit anti-β-Catenin antibody (1:200, 8480S, CST), and rabbit anti-vimentin antibody (1:500, ab137321, Abcam, Cambridge, UK).

### 2.11. Statistical Analysis

The experiment data were analyzed by GraphPad Prism 8.0 and presented as means ± SD. For normally distributed data, Student’s *t*-test was used to compare the difference between the two groups. ANOVA analysis and Tukey’s Honestly Significant Difference test were used to compare multiple groups. Brown–Forsythe or Welch tests were used to analyze heterogeneity variance data. The Mann–Whitney U-test was used to compare non-normally distributed data. *p* < 0.05 was considered for statistical significance.

## 3. Results

### 3.1. PRAME Was Differentially Expressed in Cervical Cancer and Normal Cervix

In order to verify the role of PRAME in cervical cancer, the expression of PRAME protein was explored in normal tissues and cervical cancer tissues via IHC. The PRAME expression in tumor tissues was higher than that in adjacent normal cervical tissues, and it was localized mainly in the nucleus (Figure 1A). Appendix A shows the weak, moderate, and strong staining of PRAME in cervical cancer tissues. Correspondingly, the IHC score of PRAME was higher in cervical cancer tissues (Figure 1B). Next, the differential expression of PRAME was explored among different cervical cancer cells (HeLa, SiHa, C33A, and CaSki cells) and squamous epithelial cells (Ect1/E6E7 cells) of the cervix via qRT-PCR and Western blotting. The results showed that the expression of PRAME mRNA (Figure 1C) and protein (Figure 1D,E) were highest in C33A cervical cancer cells compared to Ect1/E6E7 normal cervical epithelial cells, followed by SiHa cervical cancer cells.

### 3.2. PRAME Expression Was Involved in the Proliferation of C33A and SiHa Cells

To ascertain the function of PRAME knockdown on the growth of cervical tumor cells, PRAME-silenced C33A and SiHa cells were constructed through two sets of shRNA sequences, and the transfection efficiency was verified via Western blotting and qRT-PCR. As illustrated in Figure 1F and Appendix A, PRAME was successfully silenced in shPRAME-1 and shPRAME-2 C33A cells and SiHa cells. Cell viability and growth were impaired in PRAME-knockdown C33A and SiHa cells compared with the control groups, as detected via the CCK-8 assay (Figure 1G) and the BrdU cell proliferation assay (Figure 1H,I), respectively. Moreover, PRAME-overexpressed C33A and SiHa cells were successfully constructed in the same way (Figure 1J and Appendix A). In keeping with the above results, cell viability and proliferation were significantly increased in PRAME-overexpressed C33A and SiHa cells (Figure 1K–M).

### 3.3. PRAME Expression Regulated G0/G1 Arrest and Apoptosis in Cervical Cancer Cells

The cell cycle distribution of PRAME-knockdown and -overexpressed cells was assessed with PI staining by using flow cytometry, and cell cycle profiles were fitted by Flowjo software. As shown in Figure 2A,B, PRAME-knockdown cells were arrested more in the G0/G1 phase than the corresponding control cells. Relatively, a lower percentage of S phase cells was in the PRAME-knockdown groups compared with that in control groups. Consistently, PRAME overexpression reduced the arrest of the G0/G1 phase in C33A and SiHa cells, and the percentage of S phase was increased in PRAME-overexpressed C33A and SiHa cells compared with controls (Figure 2C,D). These revealed that PRAME-knockdown interfered with the progression of the cell cycle, while the overexpression of PRAME reduced the G0/G1 phase arrest.

In order to evaluate the effect of PRAME on cervical cancer cell apoptosis, PRAME-knockdown and -overexpressed C33A and SiHa cells were stained with PE and 7-AAD to detect the percentage of apoptotic cells. Flow cytometry revealed that the proportion of apoptotic cells was increased when PRAME was downregulated in C33A and SiHa cells (Figure 3A,B), while it was reduced when PRAME expression was upregulated (Figure 3C,D).

### 3.4. PRAME Expression Regulated Migration and Invasion of Cervical Cancer Cells

In order to investigate the effect of PRAME expression on cell migration and invasion, the scratch and transwell methods were performed. Compared with the control C33A and SiHa cells, the rate of wound recovery was attenuated in PRAME-silenced cells (Figure 4A–D). The wound-recovering ability was accordingly enhanced in C33A and SiHa cells with PRAME upregulation (Figure 4E–H). Furthermore, the migratory and invasive abilities were also detected through transwell assay. As Figure 4I,J shows, the migratory and invasive abilities of PRAME-knockdown cells were diminished in contrast with the shGFP groups, whereas the migratory and invasive abilities were boosted when the expression of PRAME was upregulated in SiHa cells (Figure 4K,L).

### 3.5. The Wnt/β-Catenin Pathway Was Regulated by PRAME Expression

As the expression of PRAME regulates migration and invasion, the EMT-related proteins E-cadherin and N-cadherin were detected via Western blotting. E-cadherin protein was upregulated, and N-cadherin was downregulated in SiHa shPRAME cells, while E-cadherin was downregulated and N-cadherin was upregulated in PRAME-overexpressed C33A cells (Figure 5A), suggesting that PRAME might affect cervical cancer metastasis by promoting EMT.

Furthermore, to illustrate the underlying mechanisms of PRAME in cervical cancer, Western blotting was employed. As displayed in Figure 5B, the expression of the key proteins of the Wnt/β-catenin signaling pathway (Wnt3a, Wnt 5a/b, p-LRP6, and β-catenin) was downregulated in PRAME-knockdown SiHa cells, while these proteins were upregulated in PRAME-overexpressed C33A cells. Moreover, the expression of the target proteins activated by Wnt/β-catenin (LEF1, CD44, and Cyclin D1) was also raised in PRAME-overexpressed C33A cells and reduced in shPRAME SiHa cells. The above results indicated that PRAME might influence the proliferation, migration, and invasion of cervical cancer cells via Wnt/β-catenin signaling pathway regulation.

To further confirm the role of PRAME in the regulation of the Wnt/β-catenin pathway, the Wnt/β-catenin inhibitor MSAB was used in PRAME-overexpressed C33A and SiHa cells and control cells. The Western blotting results showed that the activity of Wnt3a and β-catenin was attenuated in C33A cells with MSAB treatment compared with the corresponding control groups without MSAB treatment. The expression of Wnt3a and β-catenin was partially recovered in C33A control cells treated with MSAB when PRAME was overexpressed. The results were also consistent in SiHa cells (Figure 5C). In addition, cell viability and apoptosis were reversed correspondingly in PRAME-overexpressed C33A cells and SiHa cells with the use of MSAB (Figure 5D,E). Furthermore, PRAME overexpression could partially counteract the inhibition of cell migration and invasion with MSAB treatment. After treatment with MSAB, the migratory ability of PRAME-overexpressed C33A and SiHa cells was weakened compared with that of cells without MSAB treatment, but it was stronger than that in C33A and SiHa control cells with MSAB treatment (Figure 6A–D). In addition, the results of the transwell assay in SiHa cells were consistent with these findings (Figure 6E,F).

### 3.6. The Effect of PRAME on Tumor Growth via Wnt/β-Catenin Pathway In Vivo

To explore the tumorigenesis function of PRAME in vivo, the subcutaneous transplanted models of nude mice were constructed. As shown in Figure 7A, the volume of the shPRAME group tumors was smaller than the volume of the controls. The growth rate and weight of xenografts in the shPRAME group were also significantly lower than that of the control group (Figure 7B,C). In addition, TUNEL and IF were performed on the tumor tissues. The shPRAME group showed less abundant green fluorescence of anti-Ki67 antibody (Figure 7D), and the shPRAME group showed stronger apoptosis by TUNEL staining (Figure 7E).

Furthermore, the Wnt pathway inhibitor MSAB was used in vivo to verify the effect of PRAME in nude tumor growth with the regulation of the Wnt/β-catenin pathway. As shown in Figure 7F,G, the tumor volume and weight of the PRAME cDNA + DMSO group was greater than that of the control cDNA + DMSO group. Importantly, the tumor volume and weight of the PRAME cDNA + MSAB group was greater than that of the control cDNA + MSAB group, while the size and volume of PRAME-overexpressed tumors with the use of the Wnt inhibitor MSAB was smaller than that of the MSAB-untreated tumors. Additionally, the MSAB partially inhibited the tumor growth-promoting effect in vivo that was caused by PRAME overexpression (Figure 7H). The TUNEL Kit and IHC staining assay were used to analyze the apoptosis level, and β-catenin, Ki67, and vimentin expression level of tumor tissues. As shown in Figure 7I, the apoptotic level of tumor cells in PRAME-overexpressed tissues with or without MSAB therapy was reduced compared with that in the corresponding control tissues. Moreover, the apoptotic level of tissues was higher in the PRAME cDNA group after MSAB treatment than that in the PRAME cDNA group with DMSO treatment. The expression level of β-catenin, Ki67, and vimentin was higher in the PRAME-overexpressed tissues compared with the corresponding control tissues. Importantly, the β-catenin, Ki67, and vimentin expression levels were lower in the PRAME cDNA + MSAB group than that in the PRAME cDNA + DMSO group. Briefly, the β-catenin was reduced after using MSAB, the Wnt inhibitor MSAB therapy reversed the suppression of TUNEL apoptosis staining, and the facilitation of Ki67 and vimentin expression occurred via PRAME overexpression. 

## 4. Discussion

PRAME, which belongs to the CTA gene family, is abnormally expressed in several tumor tissues [8,19]. Our study demonstrated that the knockdown of PRAME inhibited cell growth and viability and accelerated cell apoptosis, while the upregulation of PRAME enhanced proliferation and abated apoptosis via the Wnt/β-catenin pathway. Moreover, cervical cancer cells were arrested more in the G0/G1 phase when PRAME was silenced, interfering with the progression of the cell cycle. The migratory and invasive potential of tumor cells is also an important marker of carcinogenesis. The scratch and transwell methods indicated the more remarkable ability of migration and invasion in PRAME-overexpressed cells. Furthermore, the effect of PRAME on cell proliferation was also confirmed in nude mice. Consistent with in vitro results, the weight and size of tumors were limited in the mice injected with PRAME-knockdown cells and greater in those injected with PRAME-overexpressed cells.

Although the role of PRAME in tumorigenesis is associated with the type of cancer, many studies have supported our conclusion. PRAME was found as a tumor-promoting gene in hematological system neoplasms [12]. Additionally, Al-Khadairi et al. [9] demonstrated that PRAME facilitated the metastasis of triple-negative breast cancer through the enhancement of the EMT process. Lee et al. [20] found that PRAME expression was associated with the malignant potential of melanoma cells. Similarly, several studies indicated that PRAME was associated with advanced stage and poor prognosis of melanoma [7,21]. Demethylation of PRAME was detected in ovarian cancer and related to a worse prognosis [22]. In addition, PRAME was more expressed in tissues of ovarian-tumor-induced death cases compared with survival cases [23]. PRAME expression was also found to be associated with the malignant potential of gastric cancer via bioinformatics data analysis [24] and was found to be a poor prognostic marker of liver cancer, promoting liver cancer cell proliferation and inhibiting cell apoptosis. Mechanistically, PRAME was found to play a role in promoting tumorigenesis by inhibiting the activation of the p53 pathway [24,25]. Bullinger at al. [26] found that PRAME facilitated cell proliferation by impairing RAR signaling. Despite this, studies regarding the effect of PRAME on cancer have focused mainly on bioinformatics analyses. The functions and specific mechanisms by which PRAME regulates cervical tumorigenesis remain unclear according to previous studies. Our study not only confirmed the enhancement of malignant biological behaviors of PRAME expression in cervical cancer cells but also involved the related activation pathways of PRAME carcinogenesis.

Dysregulation of Wnt/β-catenin signaling has been shown to be involved in the development of malignancies [27]. Activated Wnt/β-catenin signaling promotes cell proliferation and tumor stem cell renewal, having a significant role in regulating the malignant phenotype [28]. β-catenin, which is a key component of the pathway, functions as an adhesion binding protein and transcriptional coregulator implicating cell adhesion [29]. Several studies have investigated how the Wnt/β-catenin pathway is involved in the occurrence and development of cervical cancer [30], enhances EMT and metastasis potential [31], and is associated with chemotherapy drug resistance [32]. Stable β-catenin is transported into the nucleus and binds to LEF/TCF transcription factors, activating the transcription of Wnt target genes and promoting cancer metastasis [33,34]. Radich et al. [35] suggested that the progression of chronic myeloid leukemia might be involved in PRAME overexpression and Wntβ-catenin pathway activation. Nadaf et al. [36] demonstrated that PRAME was significantly upregulated in pituitary blastoma with the dysregulation of the Wnt/β-catenin signaling. However, it has not been reported that PRAME regulated tumorigenesis through the Wnt/β-catenin pathway in cervical cancer. In our study, the Wnt/β-catenin pathway was found to be activated by PRAME overexpression. The Wnt3a, Wnt5a/b, p-LRP6, and β-catenin proteins were highly expressed in PRAME-overexpressed C33A cells and downregulated in SiHa cells after PRAME silencing. In addition, the Wnt/β-catenin activated targets, LEF1, CD44, and Cyclin D1, were also positively affected by PRAME expression. Cyclin D1 is a protein associated with cell cycle regulation and acts as a regulator of cyclin-dependent kinases. The amplification or overexpression of Cyclin D1 alters cell cycle progression and leads to tumorigenesis [37]. CD44 is involved in the process of heterogeneous adhesion of tumor cells and plays a role in promoting the invasion and metastasis of tumor cells [38]. Additionally, the intracellular domain of E-cadherin interacts with β-catenin, maintaining cell adhesion [39], and it was found that in PRAME-knockdown cells, E-cadherin expression was upregulated, and N-cadherin expression was downregulated; whereas in PRAME-overexpressed cells, E-cadherin expression was decreased, and N-cadherin expression was increased. The abnormally activated Wnt/β-catenin promotes the EMT process. PRAME may participate in the EMT process through the regulation of E-cadherin, N-cadherin, and β-catenin, thereby driving tumor metastasis.

To further investigate the effect of PRAME on regulating the Wnt/β-catenin pathway, MSAB was performed to inhibit the activation of the Wnt/β-catenin signaling pathway. The use of MSAB inhibited the upregulation of Wnt3a and β-catenin caused by PRAME overexpression. The cell proliferation and apoptosis level induced by PRAME overexpression were partly reversed in C33A and SiHa cells with the use of MSAB. Consistently, MSAB also partially reversed the enhancement of migratory and invasive abilities induced by PRAME overexpression in C33A and SiHa cells. MSAB has also been used in animal experiments; therapy with Wnt inhibitor MSAB partially inhibited the tumor-growth-promoting effect in vivo caused by PRAME overexpression. The IHC and TUNEL results indicated that PRAME promoted tumor growth in vivo via the Wnt/β-catenin pathway with the enhancement of cell proliferation and EMT and the inhibition of apoptosis. Taken together, PRAME was found to have a tumor-promoting role in cervical cancer through the Wnt/β-catenin pathway.

In this study, the carcinogenesis potential of PRAME expression in cervical cancer cells was firstly investigated in vitro and in vivo. In brief, the Wnt/β-catenin pathway might be a pivotal driver of the PRAME carcinogenic effect, which could provide novel insights for the future treatment of cervical cancer and the search for new biomarkers. However, this research still has many limitations. In addition to using proliferation, apoptosis, and Wnt pathway regulation to explain the tumor-promoting effect of PRAME, the targets that directly connect between PRAME and the Wnt/β-catenin signaling pathway and the detailed mechanisms thereof need to be clarified. Additionally, it is necessary to enrich the samples of clinical cervical cancer tissues and adjacent cancerous tissues. These samples were used to further analyze the relationship between PRAME expression and clinicopathologic features in cervical cancer patients. Moreover, the models of transgenic mice could better explore the in vivo function of PRAME in cervical cancer.

## 5. Conclusions

In conclusion, PRAME was highly expressed in cervical cancer tissues and cells. PRAME overexpression expedited cell proliferation, migration, and invasion; reduced cells arrested in the G0/G1 phase; and decreased the level of cell apoptosis by the activation of the Wnt/β-catenin pathway with wnt3a, wnt5a/b, p-LRP6, β-catenin, LEF1, CD44, and Cyclin D1 upregulation. In short, PRAME was found to facilitate cervical tumorigenesis and development via the Wnt/β-catenin signaling pathway.

## Figures and Tables

**Figure 1 cancers-15-01801-f001:**
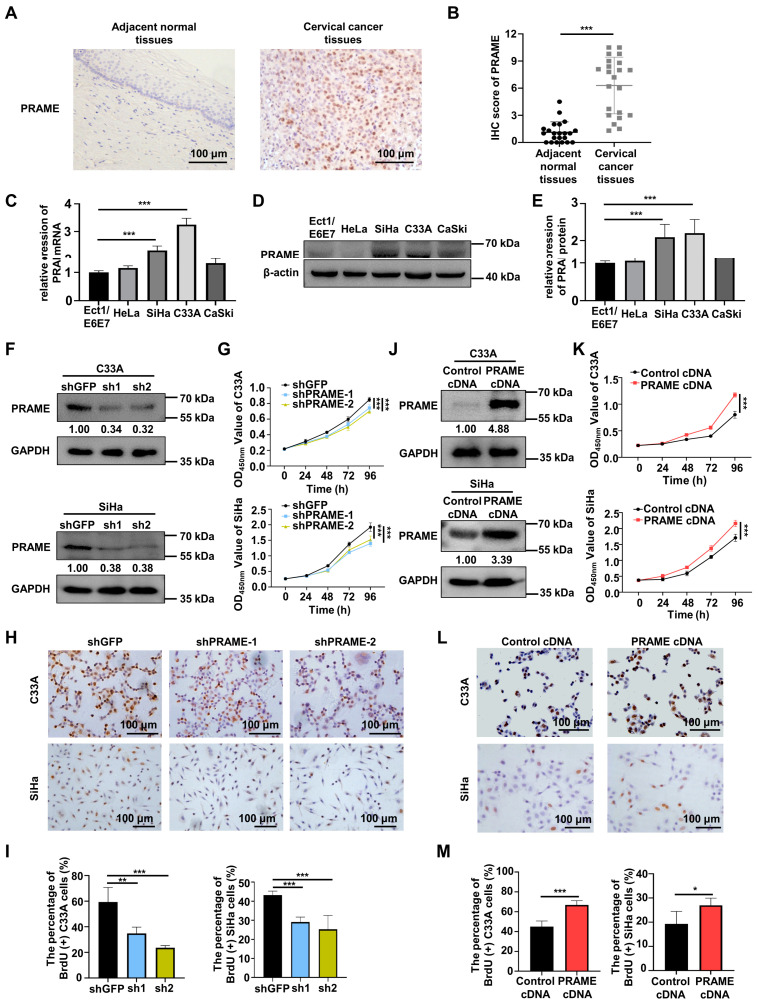
Differential expression of PRAME in cervical cancer and normal cervix, and the role of PRAME expression in cell viability and proliferation in cervical cancer cells. (**A**) The expression of PRAME protein in adjacent normal tissues and cervical cancer tissues as shown by IHC staining. (**B**) The IHC score of PRAME protein. (Adjacent normal tissues: 22 cases, Cervical cancer tissues: 22 cases). (**C**) The expression of PRAME mRNA in normal cervical epithelium cells and cervical cancer cells as shown by qRT-PCR. (**D**) The expression of PRAME protein in normal cervical epithelium cells and cervical cancer cells as shown by Western blotting. (**E**) The quantitative analysis of (**D**). (**F**) PRAME-silenced C33A and SiHa cells were constructed via the two shPRAME sequences and confirmed by Western blotting. (**G**) The effect of PRAME knockdown on cell viability in C33A and SiHa cells was detected via CCK-8 assay. (**H**) The effect of PRAME knockdown on cell proliferation in C33A and SiHa cells was measured via BrdU assay. (**I**) The quantitative results of (**H**). (**J**) PRAME-overexpressed C33A and SiHa cells were constructed and confirmed via Western blotting. (**K**) The role of PRAME overexpression in cell viability in C33A and SiHa cells was detected via CCK-8 assay. (**L**) The role of PRAME overexpression on cell proliferation in C33A and SiHa cells was measured via BrdU assay. (**M**) The quantitative results for (**L**). * *p* < 0.05, ** *p* < 0.01, *** *p* < 0.001.

**Figure 2 cancers-15-01801-f002:**
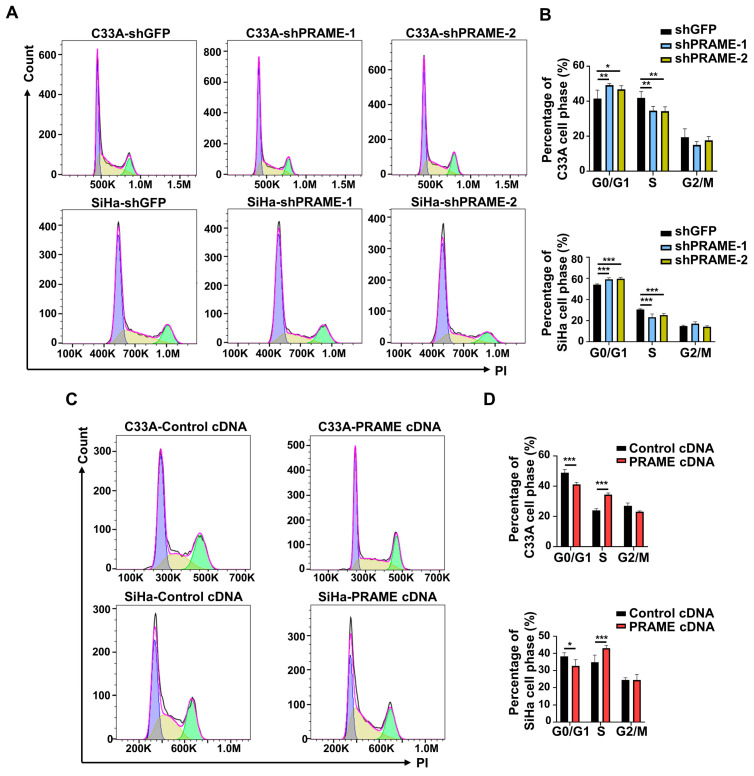
The effect of PRAME expression on cell cycle distribution of C33A and SiHa cells. The cell cycle profiles were analyzed and exported by FlowJo. The bimodal images represented the proportion of cells at different phases of the cell cycle distribution, with purple peaks representing the cells in G0/G1 phase, green peaks representing the cells in G2/M phase, and the cells in S phase labeled yellow. (**A**) The cell cycle profiles in PRAME-silenced C33A and SiHa cells. (**B**) The quantitative results for (**A**). (**C**) The cell cycle profiles in PRAME-overexpressed C33A and SiHa cells. (**D**) The quantitative results for (**C**). * *p* < 0.05, ** *p* < 0.01, *** *p* < 0.001.

**Figure 3 cancers-15-01801-f003:**
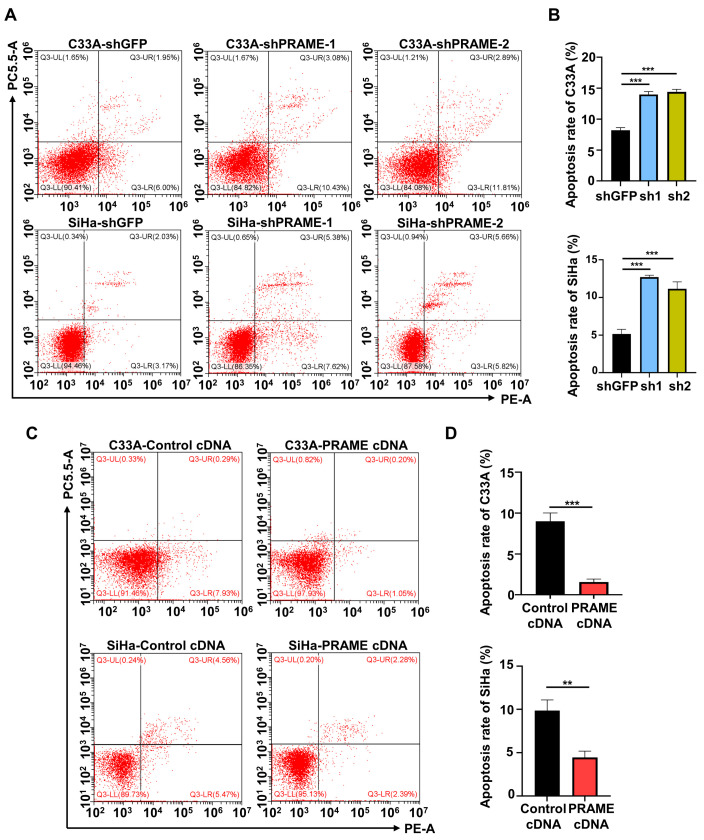
The effect of PRAME expression on cell apoptosis. The sum of cells in the upper right quadrant (UR, late apoptotic cells) and the lower right quadrant (LR, early apoptotic cells) was considered to be the apoptotic population. (**A**) The percentage of apoptotic cells in PRAME-silenced C33A and SiHa cells. (**B**) The quantitative results for (**A**). (**C**) The percentage of apoptotic cells in PRAME-overexpressed C33A and SiHa cells. (**D**) The quantitative results for (**C**). ** *p* < 0.01, *** *p* < 0.001.

**Figure 4 cancers-15-01801-f004:**
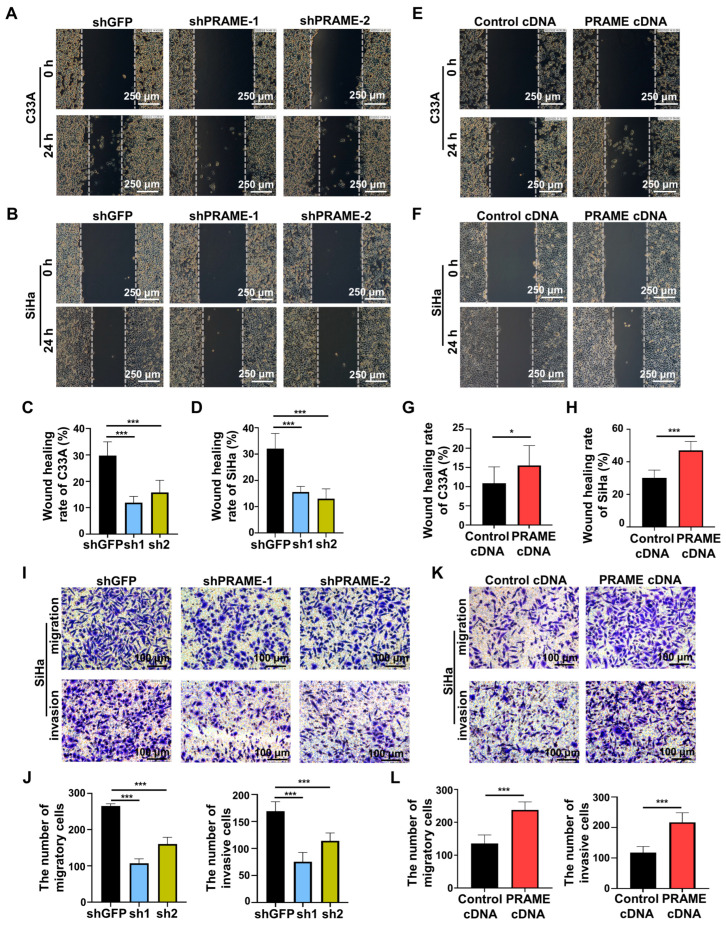
The role of PRAME expression on cell migration and invasion (**A**,**B**). The wound-healing rate in PRAME-silenced C33A and SiHa cells. (**C**,**D**) The quantitative results for (**A**,**B**). (**E**,**F**) The wound-healing rate in PRAME-overexpressed C33A and SiHa cells. (**G**,**H**) The quantitative results for (**E**,**F**). (**I**) The number of migratory and invasive cells in PRAME-knockdown SiHa cells. (**J**) The quantitative results for (**I**). (**K**) The number of migratory and invasive cells in PRAME-overexpressed SiHa cells. (**L**) The quantitative results for (**K**). * *p* < 0.05, *** *p* < 0.001.

**Figure 5 cancers-15-01801-f005:**
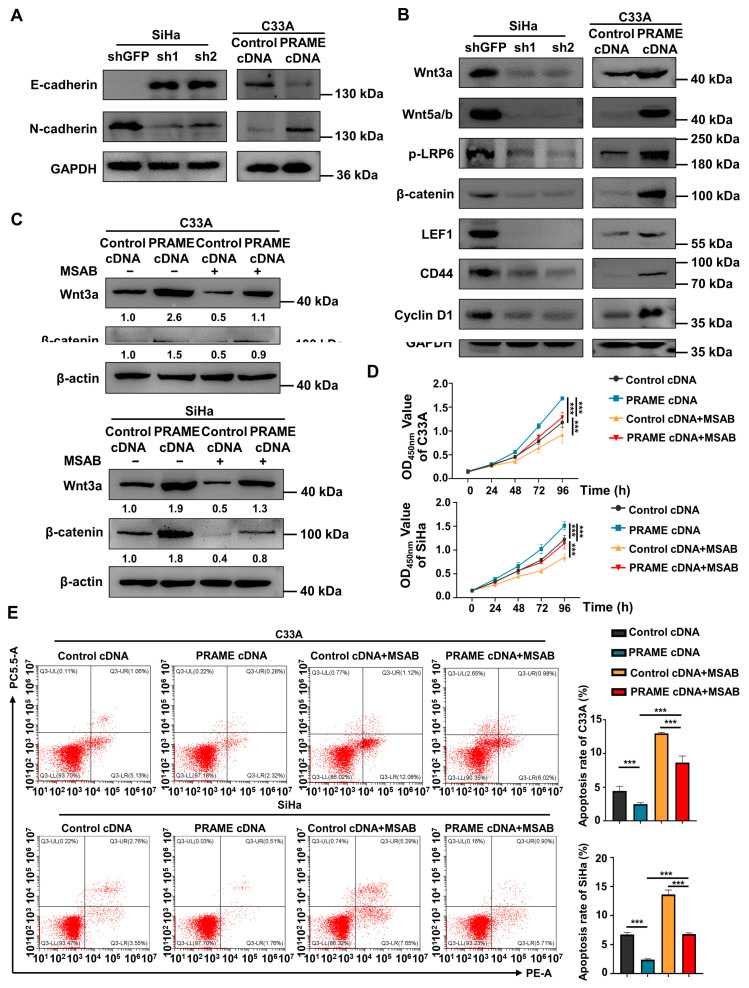
Regulation of EMT-related proteins and Wnt/β-catenin pathway by PRAME expression in cervical cancer cells. (**A**) The change in E-cadherin and N-cadherin expression in PRAME-knockdown SiHa and PRAME-overexpressed C33A cells. (**B**) Western blotting analysis of Wnt/β-catenin signaling pathway-related protein expression after PRAME knockdown in SiHa cells and PRAME overexpression in C33A cells. (**C**) Western blotting analysis of Wnt3a and β-catenin expression in PRAME-overexpressed C33A and SiHa cells after MSAB treatment. The results of gray value quantification are displayed below the corresponding bands. (**D**) The cell viability in PRAME-overexpressed and control C33A and SiHa cells with MSAB treatment. (**E**) Left panel: The cell apoptosis in PRAME-overexpressed and control C33A and SiHa cells with MSAB treatment. Right panel: The quantitative results. *** *p* < 0.001.

**Figure 6 cancers-15-01801-f006:**
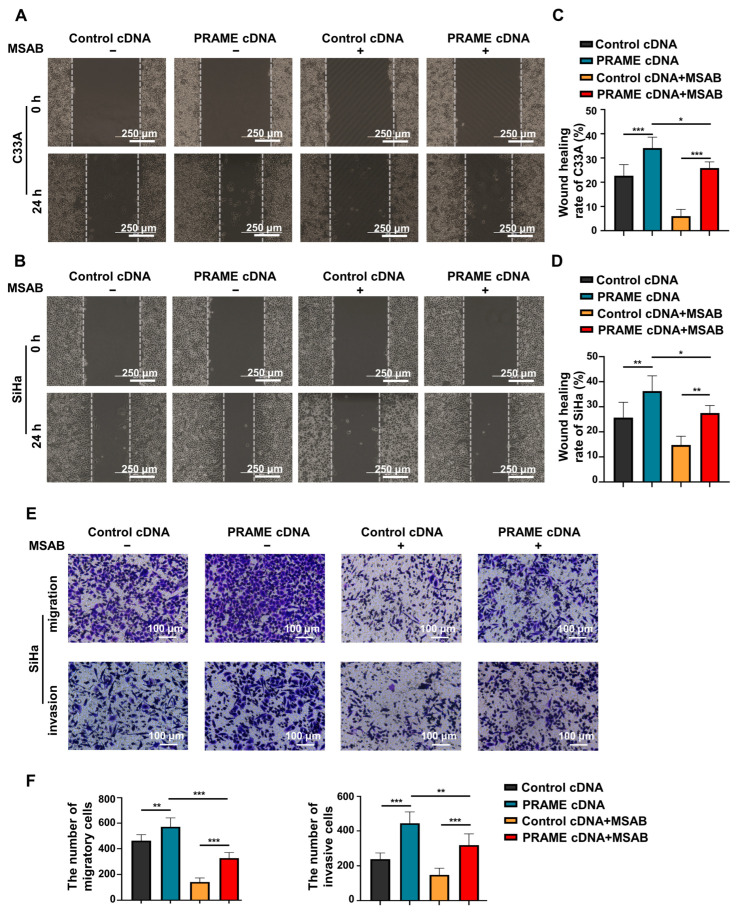
The change in migratory and invasive abilities of PRAME-overexpressed C33A and SiHa cells after MSAB treatment. (**A**,**B**) The wound-healing rate of control C33A and SiHa cells as well as PRAME-overexpressed C33A and SiHa cells after treatment with or without MSAB. (**C**) The quantitative results for (**A**). (**D**) The quantitative results for (**B**). (**E**) The number of migratory and invasive cells in control SiHa groups, PRAME-overexpressed SiHa groups, control SiHa groups with MSAB treatment, and PRAME-overexpressed SiHa groups with MSAB treatment. (**F**) The quantitative results for (**E**). * *p* < 0.05, ** *p* < 0.01, *** *p* < 0.001.

**Figure 7 cancers-15-01801-f007:**
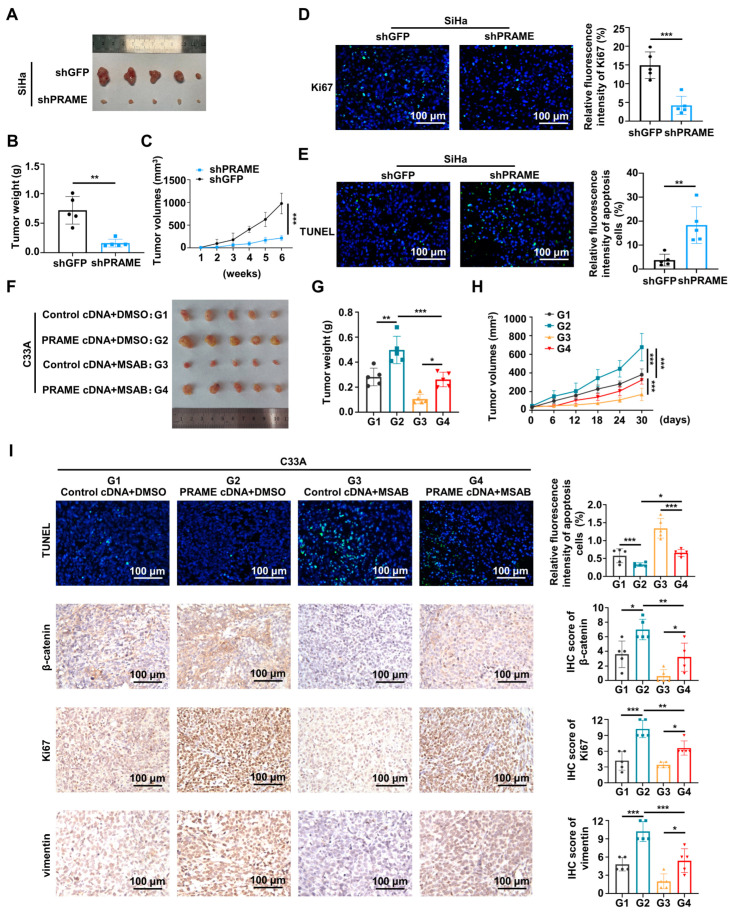
The effect of PRAME on tumor growth via Wnt/β-catenin pathway in vivo. (**A**) The image of xenografts injected with PRAME-knockdown SiHa cells and control SiHa cells. (**B**) The weight of the xenografts in PRAME-knockdown group and control group. (**C**) The growth curves of xenograft tumors of PRAME-knockdown group and control group. (**D**) The immunofluorescence staining of Ki67 in tumor slices. (**E**) The immunofluorescence staining of TUNEL assay in tumor slices. (**F**) The image of xenografts injected with PRAME-overexpressed C33A cells and control C33A cells after the MSAB therapy. (Tumor groups as G1: Control cDNA+DMSO, G2: PRAME cDNA+DMSO, G3: Control cDNA+MSAB, and G4: PRAME cDNA+MSAB). (**G**) The weight of the xenografts in PRAME-overexpressed group and control group with or without MSAB therapy. (**H**) The growth curves of the xenografts in PRAME-overexpressed group and control group with or without MSAB therapy. (**I**) The TUNEL and IHC staining of tumor tissues. * *p* < 0.05, ** *p* < 0.01, *** *p* < 0.001.

## Data Availability

All the data will be available upon request.

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
