# Peer review of "PRAME Promotes Cervical Cancer Proliferation and Migration via Wnt/β-Catenin Pathway Regulation"

_cancers, 2023, doi:10.3390/cancers15061801_

Round 1

Reviewer 1 Report (Previous Reviewer 3)

The resubmitted manuscript by Chen et al has been improved with respect to the previous submission. The authors should be commended for the effort. This version has some areas that needs to be taken care before this can be accepted. 

1. Even when the authors say that the manuscript has been corrected by a native English speaker, there still seems to lot of areas that needs extensive correction. The manuscript has places that needs to be fixed for spelling errors and grammatical mistakes: Eg: Lines 27, 68- immunology therapy to immunotherapy, line 33, line 59, line 76, 82, 126, 154, 168,  170, 180-181, 229. 297, 344, 351 (Cadherin) ,  . Lines 62-65: rewrite as the idea has not come through. Rephrase lines 132-133. Rephrase lines 146-149. Rewrite lines 191-194, 398-399, 444, 511, 521, 547.

2. Please make the abstract more concise.

3. Section 2.2: Lines 115-119. Is this an established method? Please provide reference. 

4. Section 2.7. Were the cells 'digested' for the assay??? Or is it trypsinized?

5. Rewrite section 2.9

6. Rewrite lines 302 -303. Silence of??

7. Rewrite lines 304-305.

8. The histograms do not match with the bar graphs or the text written in the results section for Fig 2. Please re-check the data provided. 

Author Response

Reviewer 2 Report (Previous Reviewer 2)

Chen et al. have significantly improved the manuscript by revising text and figures. However, some minor revision needs to be made before acceptance.

1. Figure 7 (above the Figure 6 legend) might be incorrectly placed, which should be placed above the legend of Figure 7, please modify it.

2. The IHC results of tumor tissues with MSAB treatment should be described detailly in results section (Figure 7I).

3. The activation the Wnt pathway and its target proteins in carcinogenesis by PRAME expression should be elaborated in discussion, which can benefit from the reference of relevant literatures.

4. The serial number of the method is incorrect, such as "2.3 Western blotting analysis" should be "2.4 Western blotting analysis". The rest should be modified accordingly.

Author Response

This manuscript is a resubmission of an earlier submission. The following is a list of the peer review reports and author responses from that submission.

Round 1

Reviewer 1 Report

The authors tried to demonstrate the role of PRAME in cervical cancer.  They have included many basic in vitro and in vivo experiment.

Some more details are needed:

1. The origins of the cell lines and declaration that these had been recently authenticated especially HeLa which had been criticised, and ECT1/E6E7 cells that requires special medium instead of the usual medium.

2. How the IHC was scored

3. Figures 1-E and i-I, the proliferation after knock-down and over expression of PRAME, respectively, was not very convincing

4. Fig 3 - The apoptotic populations in the Flow pictures were actually very low.  Same for the invasion assay in Fig. 4.

5. Justification of using 5 uM of MSAB in cervical cancer cell lines

6. Had the authors performed any RNAseq to see whether there was any change after MSAB?

7. Had the authors used MSAB in mice?

Author Response

Response to Reviewer 1 Comments

Dear Reviewer,

We would like to thank you for your excellent comments for our manuscript entitled “PRAME promotes cervical cancer proliferation and migration by Wnt/β-catenin pathway” (cancers-2050133). As detailed in this point-by-point response letter, we have fully addressed all the raised concerns in a satisfactory manner in our revised manuscript. Now, I am writing to submit our revised manuscript and hope that, with these extensive modifications and improvements, this manuscript would be accepted for publication in Cancers.

For convenience, the summary of the changes and answers to reviewers’ comments are listed below, and answers are marked in red. We also highlight the major changes under “track change” features in the revised revision.

Yours sincerely,

Xueqiong Zhu

Point 1: The origins of the cell lines and declaration that these had been recently authenticated especially HeLa which had been criticized, and ECT1/E6E7 cells that requires special medium instead of the usual medium.

Response 1: Thanks for your careful suggestion. The HeLa cell line used in our experiments was purchased from National Collection of Authenticated Cell Cultures (Shanghai, China), Dulbecco's Modified Eagle Medium containing 10% fetal bovine serum was used for cell culture according to the corresponding instructions of HeLa. The Ect1/E6E7 cell was purchased from Shanghai BinSui Biological Technology Co., Ltd. (China), and cultured with 90% Eagle’s Minimum Essential Medium and 10% fetal bovine serum. Both cell lines were confirmed by STR identification, and then were used in subsequent experiments. We have added the culture condition in the method section (Page 2). In addition, the STR identification reports have been uploaded in attachment.

Point 2: How the IHC was scored

Response 2: Thanks for this good suggestion. Briefly, IHC results were scored semi-quantitatively by multiplying staining intensity and staining area. Staining intensity score was classified into 0 (no staining), 1 (weak staining), 2 (moderate staining), and 3 (strong staining). Staining area score of each sample was graded as follows: 0, <5%; 1, 6–25%; 2, 26–50%; 3, 51–75%; 4, >75%. We have added the scoring and statistical methods for the immunohistochemistry assay in the methods section (Page 3).

Point 3: Figures 1-E and i-I, the proliferation after knock-down and over expression of PRAME, respectively, was not very convincing

Response 3: Thanks for this valuable comment. Bromodeoxyuridine (BrdU) assay is a method to evaluate cell proliferative ability by directly measuring the number of cells undergoing division. BrdU can be inserted into the replicated DNA double strand instead of thymidine, and the ability can be carried into progeny cells. Immunohistochemical and immunofluorescent method are generally used to visualize the BrdU content in DNA. The cell proliferation was evaluated by the percentage of BrdU positive stained cells in various studies (Cancer Sci, 2022;113(4):1250-1263; Front Oncol, 2022;12:889238; Onco Targets Ther, 2020;13:2385-2397; Bioengineered, 2021;12(1):9046-9057.). In addition, the BrdU method used to detect the cell proliferation were repeated for three times in C33A shPRMAE and PRAME overexpression cells compared with the corresponding control cells, which ensured the feasibility of our results.

Point 4: Fig 3 - The apoptotic populations in the Flow pictures were actually very low.  Same for the invasion assay in Fig. 4

Response 4: Thanks for this valuable comment. One hallmark of cancer is intrinsic or acquired resistance to apoptosis, and the apoptosis level of cancer cells is lower than that of normal cells. Several studies have shown that the apoptotic rate of cervical cancer cells without treatment was less than 5% (Cancers (Basel), 2022;14(8):1872; Mol Carcinog, 2016;55(5):918-28.). According for invasion assay, different fields of transwell membrane were randomly captured under the microscope, and the number of cells in these fields was counted and analyzed (Cancers (Basel), 2022;14(15):3694; Cancers (Basel), 2020;12(1):174.). The number of cells crossed the membrane was related to the number of cells initially inoculated, the type pf cells and the magnification of microscope. The invasive ability of cells was evaluated according to the number of cells passing through the membrane and matrigel. With the same number of inoculated cells and the magnification of microscope, the invasive ability of PRAME knockdown cells were diminished compared with that of the shGFP cells. Consistently, the migratory and invasive cells were increased when the expression of PRAME was upregulated in C33A and SiHa cells.

Point 5: Justification of using 5 uM of MSAB in cervical cancer cell lines

Response 5: We appreciate the reviewer for this excellent comment. MSAB is an inhibitor of Wnt/β-catenin pathway. A study confirmed that cell viability of Wnt-dependent cells was decreased with the treatment of MSAB (2-10 μM) while showing little effect on Wnt-independent cells and normal human cells (Cell Rep, 2016;16(1):28-36.). After referencing to some literatures, 5 μM MSAB was used to inhibit Wnt/β-catenin pathway for drug experiments in the present research (Stem Cells Int, 2021;2021:5660927; Clin Transl Med, 2022;12(2):e684; Front Cell Dev Biol, 2021;9:648201.).

Point 6: Had the authors performed any RNAseq to see whether there was any change after MSAB?

Response 6: Thank you for this careful suggestion. Indeed, this study did not perform any RNA-seq experiments to find the changes that occurred in the cells after MSAB treatment. However, according to the results of Western blotting, the expression of Wnt3a and β-catenin were partially recovered in C33A and SiHa cells treated with MSAB after PRAME overexpression, suggesting Wnt/β-catenin pathway might be a driving factor of PRAME carcinogenesis. According to your nice suggestions, RNA-seq experiments will be performed to further confirm the role of Wnt/β-catenin pathway in transcriptional level on PRAME regulating cervical cancer occurrence and development in our future study.

Point 7: Had the authors used MSAB in mice

Response 7: We appreciate the reviewer for this excellent comment. Indeed, this study did not perform any in vivo experiments with the use of MSAB. However, in order to explore the specific signaling pathways of PRMAE on carcinogenesis, the MSAB were performed in vitro through Western blotting and biological function experiments. Mechanistically, PRAME was found to promote cervical tumorigenesis and development via the Wnt/β-catenin signaling pathway and the process of epithelial–mesenchymal transition in cercial cancer. In our future research, the animal experiments with the use of MSAB will be perform to further verify the effect of Wnt/β-catenin pathway in PRAME carcinogenesis and explore the targets directly connecting between PRAME and the Wnt/β-catenin signaling pathway.

Reviewer 2 Report

In this manuscript, Chen et al. have explored the role of PRAME in regulating cervical cancer growth and metastasis. Based on a series of gain-of-function and loss-of-function experiments, the authors found that PRAME regulated cell proliferation, migration, invasion, apoptosis and cell cycle through the Wnt/β-catenin pathway. Furthermore, the results were confirmed in vivo using xenografts models. Though the experiment of pathway inhibitor was a plus point of the manuscript, some points in the manuscript should be addressed, and the manuscript will be accepted after minor revision.

1.     Scratch assay was used to show the effect of PRAME on cell migratory ability but the authors failed to explain how to avoid interference from cell proliferation in the progress.

2.     The image scales of scratch assay and transwell assay results were inconsistent between fig. 4 and fig. 6. Could the image field be unified? Besides, the immunofluorescence results in Figure 7D did not indicate magnification.

3.     CCK-8 was performed on PRAME-overexpressed C33A and SiHa cells after the use of the MSAB, however, the color identification of curve of group control cDNA+MSAB and PRMAE cDNA+MSAB was not clear.

4.     The author mentioned that PRAME expression could partially reverse the effect of Wnt/β-catenin pathway inhibitor by western blotting, quantization of gray value and normalization of internal parameters would make the results more visual.

5.     Authors did not explain in detail how PRAME regulated Wnt/β-catenin pathway promote tumor proliferation, invasion and migration, the study would benefit from the reference and discussion of relevant literatures.

6.     There are some errors in the manuscript, language need to be corrected for scientific accuracy by a native English speaker.

Author Response

Response to Reviewer 2 Comments

Dear Reviewer,

We would like to thank you for your excellent comments for our manuscript entitled “PRAME promotes cervical cancer proliferation and migration by Wnt/β-catenin pathway” (cancers-2050133). As detailed in this point-by-point response letter, we have fully addressed all the raised concerns in a satisfactory manner in our revised manuscript. Now, I am writing to submit our revised manuscript and hope that, with these extensive modifications and improvements, this manuscript would be accepted for publication in Cancers.

For convenience, the summary of the changes and answers to reviewers’ comments are listed below, and answers are marked in red. We also highlight the major changes under “track change” features in the revised revision.

Yours sincerely,

Xueqiong Zhu

Point 1: Scratch assay was used to show the effect of PRAME on cell migratory ability but the authors failed to explain how to avoid interference from cell proliferation in the progress.

Response 1: Thanks for your careful suggestion. Initially, the cells were cultured in a complete medium containing fetal bovine serum, and a scratch was created using a 10 μL pipette tip after the confluence of cells approached 100%. Then, the cells were cultured in a serum-free medium and imaged no more than 24 hours later to avoid the effect of proliferation. The migratory ability was evaluated through the wound healing rate. We have added these details in the methods (Page 4).

Point 2: The image scales of scratch assay and transwell assay results were inconsistent between fig. 4 and fig. 6. Could the image field be unified? Besides, the immunofluorescence results in Figure 7D did not indicate magnification.

Response 2: We greatly thank the reviewer for these positive and insightful comments. In the revised Figure 4 and Figure 6, the image scales of scratch and transwell assay results were maintained in a consistent manner. In addition, the image scales were also added in the immunofluorescence results in the revised Figure 7D.

Point 3: CCK-8 was performed on PRAME-overexpressed C33A and SiHa cells after the use of the MSAB, however, the color identification of curve of group control cDNA+MSAB and PRMAE cDNA+MSAB was not clear

Response 3: Thanks for your careful suggestion. In the revised manuscript, a bolder color scheme has been used both in line and bar graphs, which could be better distinguished Control cDNA, PRAME cDNA, Control cDNA + MSAB and PRAME cDNA + MSAB groups (revised Figure 5 and revised Figure 6).

Point 4: The author mentioned that PRAME expression could partially reverse the effect of Wnt/β-catenin pathway inhibitor by western blotting, quantization of gray value and normalization of internal parameters would make the results more visual.

Response 4: We thank the reviewer for this good suggestion. In the revised Figure 5C, the expression of Wnt3a and β-catenin proteins have been quantified by gray value analysis, and the values of the internal parameter normalization have been changed below the corresponding bands, which made the results more visual.

Point 5: Authors did not explain in detail how PRAME regulated Wnt/β-catenin pathway promote tumor proliferation, invasion and migration, the study would benefit from the reference and discussion of relevant literatures.

Response 5: We greatly thank the reviewer for these positive and insightful comments. By referring to several studies, we have rediscussed more about the way PRAME was involved in Wnt/β-catenin, promoting cell proliferation, migration, invasion and process of epithelial–mesenchymal transition (page 14, paragraph 3 and 4 of the discussion section). However, there has not been reported that PRAME regulated tumorigenesis through Wnt/β-catenin pathway in cervical cancer. In the present study, we firstly demonstrated that the Wnt/β-catenin pathway might be a pivotal driver of PRAME carcinogenic effect in cervical cancer. Furthermore, we will explore the directly acting proteins of PRAME in activating the Wnt/β-catenin pathway in future study.

Point 6: There are some errors in the manuscript, language need to be corrected for scientific accuracy by a native English speaker.

Response 6: Thank you for your careful suggestion. The manuscript has been revised by a native English speaker from MDPI Author Services. We have also carefully checked the manuscript to minimize typographical, grammatical, and bibliographical errors before submitting.

Reviewer 3 Report

The work described by Chen et al addresses the role of PRAME in cervical cancer through the Wnt signaling pathway. The research follows a predictable and traditional approach in tackling the question. 

The language used through out the manuscript needs extensive editing.

Suppl Fig 1 shows HeLa showing lesser expression for PRAME - lesser than Ect1 - what could be the factor that controls the differential expression of PRAME in cervical cancer cell lines?

Fig1: what was the knock down and knock in efficiency that was obtained? And does this correlate with the effect on cell proliferation shown?

Fig 2A : When comparing please use the same scale in Y axis - Check C33AshPRAME1. Similar error in Fig 2C.

For the Annexin/PI staining done for apoptosis assay - please mention the quadrants considered as apoptotic population.

Fig 5D: For the line graph, please use a different color scheme, as it is difficult to comprehend.

A model figure summarizing the findings would add value to the paper. 

Author Response

Response to Reviewer 3 Comments

Dear Reviewer,

We would like to thank you for your excellent comments for our manuscript entitled “PRAME promotes cervical cancer proliferation and migration by Wnt/β-catenin pathway” (cancers-2050133). As detailed in this point-by-point response letter, we have fully addressed all the raised concerns in a satisfactory manner in our revised manuscript. Now, I am writing to submit our revised manuscript and hope that, with these extensive modifications and improvements, this manuscript would be accepted for publication in Cancers.

For convenience, the summary of the changes and answers to reviewers’ comments are listed below, and answers are marked in red. We also highlight the major changes under “track change” features in the revised revision.

Yours sincerely,

Xueqiong Zhu

Point 1: The work described by Chen et al addresses the role of PRAME in cervical cancer through the Wnt signaling pathway. The research follows a predictable and traditional approach in tackling the question. The language used throughout the manuscript needs extensive editing.

Response 1: We greatly thank the reviewer for these positive and insightful comments. Following these outstanding instructions, we have extensively modified our manuscript accordingly. The manuscript has been revised by a native English speaker from MDPI Author Services before resubmitting. We have also carefully checked the manuscript to minimize typographical, grammatical, and bibliographical errors before submitting.

Point 2: Suppl Fig 1 shows HeLa showing lesser expression for PRAME - lesser than Ect1 - what could be the factor that controls the differential expression of PRAME in cervical cancer cell lines?

Response 2: Thanks for your careful suggestion. The expression of PRAME in normal cervical cell and cervical cancer cells were quantitatively analyzed by gray value, and was normalized with the expression of internal reference GAPDH. The quantitative result has been presented in lower panel of revised Figure 1C after repeated experiments for three times. The expression of PRAME in HeLa was higher than that in Ect1/E6E7. A number of studies have shown that the differential expression of genes in different cervical cancer cell lines might be related to the origin of the cells, additionally, it might also be related to culture conditions, and the type and status of HPV infection (Clin Cancer Res, 2013;19(5):1197-203; Epigenetics, 2011;6(6):777-87; Front Immunol, 2022;13:801639.). Moreover, the cause of the differential expression of PRAME gene in cervical cancer cells and its carcinogenic effect needs to be further studied in the future.

Point 3: Fig1: what was the knock down and knock in efficiency that was obtained? And does this correlate with the effect on cell proliferation shown?

Response 3: We appreciate the reviewer for this excellent comment. Transfection efficiency was analyzed via Western botting. The expression of PRAME protein in cells transfected shGFP, shPRAME-1 and shPRAME-2 (control cDNA and PRAME cDNA) have been quantified through gray value analysis by Image J and the gray values have been presented below the bands. The knockdown efficiencies of PRAME were respectively 65.7% and 67.6% in shPRAME-1 and shPRAME-2 C33A cells, and the knockdown efficiencies of PRAME were respectively 62.3% and 61.8% in shPRAME-1 and shPRAME-2 SiHa cells. The overexpressed efficiency of PRAME in C33A cells was 380% and the overexpressed efficiency of PRAME in SiHa cells was 239%. Generally, the transfection can be considered successful when the knockdown efficiency is higher than 50% (Nat Cell Biol, 2014;16(1):10-8; BMC Biotechnol, 2012;12:42; Anal Biochem, 2011;417(1):162-4. Cancers (Basel), 2020;12(10):2757.). If the gene is related to proliferation or apoptosis, the higher the knockdown efficiency, the greater the impact on cancer cell biological behaviors such as cell proliferation and apoptosis abilities (Cancer Res, 2016;76(20):6054-6065; Sci Rep, 2021;11(1):19799; Osteoarthritis Cartilage, 2007;15(11):1275-82.). In the present research, the knockdown efficiency of PRAME was greater than 50%, and the overexpression efficiency was greater than 200%, considering to be the successful construction of PRAME knockdown and overexpression cell lines, which made sense to use these cells for follow-up experiments.

Point 4: When comparing please use the same scale in Y axis - Check C33AshPRAME1. Similar error in Fig 2C.

Response 4: Thanks for your careful suggestion. However, as several studies have shown that the cell cycle profiles were analyzed and exported by FlowJo software, and the scale of Y-axis of the exported images from the software were fixed and unable to adjust (Cancer Commun (Lond), 2021;41(6):492-510; Cell Death Dis, 2016;7(10):e2409.). The percentage of cells at different cell cycle stages was compared, the quantitative results were showed in the revised Figure 2B and Figure 2D.

Point 5: For the Annexin/PI staining done for apoptosis assay - please mention the quadrants considered as apoptotic population.

Response 5: We appreciate the reviewer for this excellent comment. In the revised legend of Figure 3, we have explained that the sum of the percentage of cells in the upper right quadrant (UR, late apoptotic cells) and the lower right quadrant (LR, early apoptotic cells) was considered as apoptotic population.

Point 6: Fig 5D: For the line graph, please use a different color scheme, as it is difficult to comprehend.

Response 6: Thanks for your careful suggestion. In the revised manuscript, a bolder color scheme has been used both in line and bar graphs, which could be better distinguished Control cDNA, PRAME cDNA, Control cDNA + MSAB and PRAME cDNA + MSAB groups (revised Figure 5 and revised Figure 6).

Point 7: A model figure summarizing the findings would add value to the paper.

Response 7: We appreciate the reviewer for this excellent comment. A model figure has been presented in revised Figure 7F and summarized findings of our research.

Reviewer 4 Report

General comments: This is an interesting study investigating a novel putative oncogene in cervical cancer. There are numerous grammatical errors that need to be fixed. I have listed some of them below. The discussion should be restructured, and the methods should contain more info about the material and methods (see details below).

Specific comments in chronological order:

Please rephrase the Simple Abstract and Abstract to make it less repetitive.

Line 24: Detected not detect

Line 78: Please explain MSAB.

Methods: You should explain better what kind of transfected cells you ended up with in the analyses (e.g. sh-PRAME-1 vs sh-PRAME-2 (why two, what is the difference) PRAME cDNA (why only one)

Line 142: Explain PI. Consider writing a description of the method here and refer to that in the results section.

Line 154: please use another word than ‘conducted’. I think ‘used’ or ‘applied’ is what you mean.

Line: 162: It is better to say ‘sacrificed’ than ‘killed’.

Line 170: ‘present’ not ‘presented’.

Line 175-176: This sentence includes several grammatical errors (please correct): Data of multiple groups were used ANOVA analysis, and Tukey' 175 s Honestly Significant Difference test was used to further multiple comparison.

Results: Please consider including Supplementary figure 1 as a main figure. Please state how many patient tumors were stained and the clinicopathological characteristics of the patients.

Line 216-217: Rephrase ‘assessment’ to ‘assessed’ and ‘shown’ to ‘shows’.

Figure 2: Please explain figure A and C better.

Line 235: Please rephrase ‘testify’ to e.g. ‘test’.

Line 244-248: I would suggest putting this sentence to the next paragraph.

Figure 5 D: The + MSAB curves are hard to see.

Discussion: Please reconstruct the discussion. The first paragraph should have some of your key findings in it and then a discussion of how this fit with existing literature. The way it is now, it belongs to the introduction. Please discuss more about the way PRAME is involved in Wnt/b-catenin, EMT and metastasis also bringing in other studies. Please elaborate more on limitations/strengths in the experimental set up.

Author Response

Response to Reviewer 1 Comments

Dear Reviewer,

We would like to thank you for your excellent comments for our manuscript entitled “PRAME promotes cervical cancer proliferation and migration by Wnt/β-catenin pathway” (cancers-2050133). As detailed in this point-by-point response letter, we have fully addressed all the raised concerns in a satisfactory manner in our revised manuscript. Now, I am writing to submit our revised manuscript and hope that, with these extensive modifications and improvements, this manuscript would be accepted for publication in Cancers.

For convenience, the summary of the changes and answers to reviewers’ comments are listed below, and answers are marked in red. We also highlight the major changes under “track change” features in the revised revision.

Yours sincerely,

Xueqiong Zhu

Point 1: General comments: This is an interesting study investigating a novel putative oncogene in cervical cancer. There are numerous grammatical errors that need to be fixed. I have listed some of them below. The discussion should be restructured, and the methods should contain more info about the material and methods (see details below).

Response 1: We greatly thank the reviewer for these positive and insightful comments. Following your outstanding instructions, we have extensively modified our manuscript accordingly.

Point 2: Please rephrase the Simple Abstract and Abstract to make it less repetitive.

Response 2: Thanks for your careful suggestion. The manuscript has been revised by a native English speaker from MDPI Author Services. Besides, we have made sure that the Simple Abstract and Abstract were revised and checked before submission to reduce statement repetition.

Point 3: Line 24: Detected not detect

Response 3: Thanks for your careful suggestion. We apologize for the grammar mistake. We have corrected this sentence as “the expression of PRAME in cervical tissues and cells was detected by the immunohistochemistry (IHC)”. (Page 1)

Point 4: Line 78: Please explain MSAB.

Response 4: Thanks for your nice suggestion. MSAB is a potent inhibitor of Wnt/β-catenin signaling, which can downregulate target genes of Wnt/β-catenin pathway. MSAB has selective antitumor effects on Wnt-dependent cancer cells. We have added these details in the method section. (Page 2)

Point 5: Methods: You should explain better what kind of transfected cells you ended up with in the analyses (e.g. sh-PRAME-1 vs sh-PRAME-2 (why two, what is the difference) PRAME cDNA (why only one)

Response 5: Thanks for this valuable comment. Due to the different principles of gene knockdown and overexpression, the methods of plasmid construction are different (BMC Biotechnol, 2006;6:7; Genetics, 2012;190(3):841-54.). For the gene overexpression experiment, the full-length sequence of the gene has been used to construct overexpressed plasmids. The PRMAE clones were linked to an overexpression vector system. For the knockdown experiment, potential target sites in mRNA were found according to the guidelines, and two to four shRNA sequences were designed for each gene depending on the different target sites to prevent off-target effects (Methods Mol Biol, 2010;629:141-58; N Am J Med Sci, 2010;2(12):598-601.). In general, the same effect of two shRNAs was required to ensure the validity of experiments. We have also added the sh-PRAME-1, sh-PRAME-2 and PRAME cDNA plasmid sequences in the section of method (page 3).

Point 6: Line 142: Explain PI. Consider writing a description of the method here and refer to that in the results section.

Response 6: Thanks for your nice suggestion. Propidium Iodide (PI) is a fluorescent nucleic acid dye, which could be selectively embedded between the bases of DNA double-stranded helix. The amount of PI binding is directly proportional to the content of DNA. The DNA distribution of each stage of the cell cycle have been analyzed by flow cytometry and Flowjo 10.0, and the percentage of each stage of the cell cycle was calculated. We have added these details in the method section (page 4), making the understanding of the results of cell cycle experiment much easier.

Point 7: Line 154: please use another word than ‘conducted’. I think ‘used’ or ‘applied’ is what you mean.

Response 7: Thanks for your nice suggestion. We have corrected the word as “applied” in the revised manuscript (page 4).

Point 8: Line: 162: It is better to say ‘sacrificed’ than ‘killed’.

Response 8: Thanks for your careful suggestion. We have corrected the writing as ‘The mice were sacrificed’ in the revised manuscript (page 4).

Point 9: Line 170: ‘present’ not ‘presented’.

Response 9: Thanks for your nice suggestion. We apologize for the grammar and typo mistakes. We have corrected the word as “present” in the revised manuscript (page 4).

Point 10: Line 175-176: This sentence includes several grammatical errors (please correct): Data of multiple groups were used ANOVA analysis, and Tukey' 175 s Honestly Significant Difference test was used to further multiple comparison.

Response 10: Thanks for your careful suggestion. We apologize for the grammar mistakes, and we have corrected the sentence as “ANOVA analysis and Tukey' s Honestly Significant Difference test were used to compare multiple groups” in the revised manuscript (page 5).

Point 11: Results: Please consider including Supplementary figure 1 as a main figure. Please state how many patient tumors were stained and the clinicopathological characteristics of the patients.

Response 11: Thanks for this valuable comment. The results of previous Supplementary Figure 1 have been presented in the revised Figure 1A-C. Sixteen patients with cervical cancer were included in our research for IHC assay. By scoring the staining of cervical cancer tissues and adjacent normal cervical tissues, we found that 16 cases of cervical cancer tissues were all stained, and the PRAME IHC score of cervical cancer tissues was higher than that of adjacent normal cervical tissues. Indeed, due to the insufficient number of cases, we did not analyze clinicopathological characteristics of the patients, which will be explored in our future research. At present, our results have indicated that the expression of PRAME in cervical cancer tissues was higher than that in adjacent normal cervical tissues.

Point 12: Line 216-217: Rephrase ‘assessment’ to ‘assessed’ and ‘shown’ to ‘shows’.

Response 12: Thanks for your nice suggestion. We apologize for the grammar and typo mistakes. We have corrected these mistakes in the revised manuscript accordingly (page 7) and have also carefully checked the manuscript to minimize typographical, grammatical, and bibliographical errors before submitting.

Point 13: Figure 2: Please explain figure A and C better.

Response 13: Thanks for your careful suggestion. Cell cycle profiles were fitted by FlowJo. The G0/G1 phase cells were marked as purple, the S phase cells were marked as yellow and G2/M phase cells were marked as red in the images. We have explained the Figure 2A and 2C in detail in revised manuscript include the part of result and figure legend.

Point 14: Line 235: Please rephrase ‘testify’ to e.g. ‘test’.

Response 14: Thanks for your nice suggestion. We have corrected the corresponding part of the revised manuscript (page 8).

Point 15: Line 244-248: I would suggest putting this sentence to the next paragraph.

Response 15: Thank you for your careful suggestion. We have corrected the description in the results section accordingly, put the description of E-cadherin and N-cadherin expression to the next paragraph (page 9).

Point 16: Figure 5 D: The + MSAB curves are hard to see.

Response 16: Thanks for your careful suggestion. In the revised manuscript, a bolder color scheme has been used both in line and bar graphs, which could be better distinguished Control cDNA, PRAME cDNA, Control cDNA + MSAB and PRAME cDNA + MSAB groups (revised figure 5 and revised figure 6).

Point 17: Discussion: Please reconstruct the discussion. The first paragraph should have some of your key findings in it and then a discussion of how this fit with existing literature. The way it is now, it belongs to the introduction. Please discuss more about the way PRAME is involved in Wnt/b-catenin, EMT and metastasis also bringing in other studies. Please elaborate more on limitations/strengths in the experimental set up.

Response 17: We greatly thank the reviewer for these positive and insightful comments. We have reconstructed the description in the discussion section accordingly to analyze more about the way PRAME is involved in Wnt/β-catenin, EMT and metastasis by referring to multiple literatures. The expression of PRAME was positively associated with metastasis in uveal melanoma (Clin Cancer Res, 2016;22(5):1234-42.), gastric cancer (Ann Surg Oncol, 2020;27(6):2071-2080.) and osteosarcoma (Biochem Biophys Res Commun, 2012;419(4):801-8.). However, there is no study confirmed that PRAME expression promoted in the occurrence and progression of cancer through Wnt/β-catenin pathway. In the present study, we firstly demonstrated that the Wnt/β-catenin pathway might be a pivotal driver of PRAME carcinogenic effect in cervical cancer. Furthermore, we will explore the directly acting proteins of PRAME in activating the Wnt/β-catenin pathway in future study. In addition, the limitations/strengths in the experimental set up were also discussed in the revised manuscript.